# A Novel 4-gene Score to Predict Survival, Distant Metastasis and Response to Neoadjuvant Therapy in Breast Cancer

**DOI:** 10.3390/cancers12051148

**Published:** 2020-05-02

**Authors:** Masanori Oshi, Eriko Katsuta, Li Yan, John M.L. Ebos, Omar M. Rashid, Ryusei Matsuyama, Itaru Endo, Kazuaki Takabe

**Affiliations:** 1Department of Surgical Oncology, Roswell Park Comprehensive Cancer Center, Buffalo, NY 14263, USA; asa1101oshi@gmail.com (M.O.); eriko.katsuta@roswellpark.org (E.K.); 2Department of Gastroenterological Surgery, Yokohama City University School of Medicine, Yokohama 236-0004, Japan; ryusei@yokohama-cu.ac.jp (R.M.); endoit@med.yokohama-cu.ac.jp (I.E.); 3Department of Biostatistics & Bioinformatics, Roswell Park Comprehensive Cancer Center, Buffalo, NY 14263, USA; li.yan@roswellpark.org; 4Department of Cancer Genetics and Genomics, Roswell Park Comprehensive Cancer Center, Buffalo, NY 14263, USA; John.Ebos@roswellpark.org; 5Department of Surgery, Holy Cross Hospital, Michael and Dianne Bienes Comprehensive Cancer Center, Fort Lauderdale, FL 33308, USA; omarmrashidmdjd@gmail.com; 6Department of Surgery, Massachusetts General Hospital, Boston, MA 02114, USA; 7Department of Surgery, University at Buffalo Jacobs School of Medicine and Biomedical Sciences, State University of New York, Buffalo, NY 14263, USA; 8Department of Breast Surgery and Oncology, Tokyo Medical University, Tokyo 160-8402, Japan; 9Department of Surgery, Niigata University Graduate School of Medical and Dental Sciences, Niigata 951-8510, Japan; 10Department of Breast Surgery, Fukushima Medical University School of Medicine, Fukushima 960-1295, Japan

**Keywords:** biomarker, breast cancer, gene expression, metastasis, prognosis, treatment response

## Abstract

We generated a 4-gene score with genes upregulated in LM2-4, a metastatic variant of MDA-MB-231 (DOK 4, HCCS, PGF, and SHCBP1) that was strongly associated with disease-free survival (DFS) in TCGA cohort (hazard ratio [HR]>1.2, *p* < 0.02). The 4-gene score correlated with overall survival of TCGA (HR = 1.44, *p* < 0.001), which was validated with DFS and disease-specific survival of METABRIC cohort. The 4-gene score was able to predict worse survival or clinically aggressive tumors, such as high Nottingham pathological grade and advanced cancer staging. High score was associated with worse survival in the hormonal receptor (HR)-positive/Her2-negative subtype. High score enriched cell proliferation-related gene sets in GSEA. The score was high in primary tumors that originated, in and metastasized to, brain and lung, and it predicted worse progression-free survival for metastatic tumors. Good tumor response to neoadjuvant chemotherapy or hormonal therapy was accompanied by score reduction. High scores were also predictive of response to neoadjuvant chemotherapy for HR-positive/Her2-negative subtype. High score tumors had increased expression of T cell exhaustion marker genes, suggesting that the score may also be a biomarker for immunotherapy response. Our novel 4-gene score with both prognostic and predictive values may, therefore, be clinically useful particularly in HR-positive breast cancer.

## 1. Introduction

Although breast cancer has a relatively better prognosis compared to other cancers, approximately 40,000 patients die annually from this disease in the US [1]. The most commonly used tool to predict patients’ prognosis has been American Joint Committee on Cancer (AJCC) Cancer Staging. Its latest (8th) edition adds biological parameters such as tumor expression status for the estrogen receptor (ER), progesterone receptor (PR), and human epidermal growth factor receptor 2 (HER2), and pathological grade to the traditional anatomical TNM staging criteria, tumor size, lymph node metastasis, and distant metastasis. Metastasis to distant organs such as the brain and lung is the most prognostic, accounting for one half to three quarters of breast cancer deaths [2]. However, to date, there is no effective means to predict which breast cancer patients will develop metastasis. 

Although treatment for metastatic breast cancer is essential to further improve breast cancer survival, currently there is no cure that can eradicate metastatic disease. On the other hand, treatment for cancer has improved significantly during the last decade, with new regimens and approaches in chemotherapy and immunotherapy bringing considerable benefits for late-stage cancer patients. For example, metastatic melanoma, once considered a fatal diagnosis, has become a curable disease to some with the advent of immunotherapy [3]. Early identification of breast cancer before it becomes metastatic may allow for successful treatment with such therapies. 

The availability of high-throughput technology that is not economically prohibitive has led to the rapid accumulation of data on transcriptome-wide gene expression of tumors from a large number of patients. Exploration of such data by the research community has been of importance to the identification of potential biomarkers that may be more accurate at the individual patient level [4]. Cancer research with genetic analysis has developed dramatically with the availability and accessibility to data collected from around the world with projects such as The Cancer Genome Atlas (TCGA) [5] and data repositories such as Gene Expression Omnibus (GEO). This research activity has been aided by developments in computational biology that today make it possible to, among other things, use gene expression profiles of tumors to quantify their enrichment of many cancer-related pathways with Gene set enrichment analysis (GSEA) [6]. These cohorts and algorithms allow us to conduct research that assesses the real-world relevance of the expression of genes of interest [7,8,9,10,11,12,13,14,15,16,17,18,19,20,21]. 

In this study, we compared the gene expression profiles of poorly metastatic human breast cancer cell line (MDA-MB-231) with a highly-metastatic variant (LM2-4) that was selected after multiple rounds of orthotopic implantation into immune compromised mice and then, following surgical removal of a primary tumor, re-selected from spontaneous metastasis in the lung [22,23]. Transcriptomic comparisons were undertaken to identify genes whose expression may affect aggressiveness and, using concurrent clinical and molecular information from a large cohort of breast cancer patients, we developed a gene signature score that was associated with survival, distant metastasis, and response to neoadjuvant therapy in multiple validation cohorts. Because of the association of the score with such a wide variety of phenotypes, we hypothesized that the score may reflect the balance of overall cancer aggressiveness with anti-cancer immune response.

## 2. Results 

### 2.1. Development of a Prognostic 4-Gene Score for Breast Cancer

The human metastatic breast cell variant LM2-4 selected in vivo from the parental MDA-MB-231 cell line has been extensively described, with studies based on organ tropic seeding to the lungs and how it can be impacted by surgery and treatment detailed elsewhere [22,23,24,25,26,27,28,29]. However, to date no study has extensively examined the molecular differences that may explain the substantial metastatic phenotype changes between these two cell variants. We hypothesized that the transcriptomic profile of LM2-4 will contain a gene expression signature that is associated with breast cancer metastasis in humans. To this end, we performed RNA sequencing of the two cell lines in triplicate to compare their transcriptomes. 

The expression of 297 genes was higher by ≥ 1.2 log_2_ units in LM2-4 compared to MDA-MB-231 with false discovery rate (FDR) < 0.05, whereas the expression was significantly reduced for 250 genes. To understand the relevance of the 297 upregulated genes for metastasis, we examined the association of their expression in treatment-naive primary tumors with disease-free survival (DFS) in the TCGA breast cancer cohort of 1093 patients. DFS was deemed an appropriate indicator of metastasis since the majority of breast cancer recurrences were as metastasis. Using univariate Cox regression analyses we identified four genes, *DOK4*, *HCCS*, *PGF* and *SHCBP1*, whose tumoral expression was significantly associated with DFS with HR > 1.2 and *p* < 0.02 (there were 5 genes with *p* = 0.02–0.05). We generated a multivariate Cox regression model for DFS using expression values of the 4 genes. A 4-gene score using tumor gene expression and Cox regression coefficient values in the model was calculated as: 1.355 × (expression*^DOK4^*) + 1.641 × (expression*^HCCS^*) + 1.345 × (expression*^PGF^*) + 1.232 × (expression*^SHCBP1^*).

### 2.2. Breast Cancer Patients with High 4-Gene Score have Worse Survival

We first confirmed the prognostic value of the 4-gene score using the TCGA cohort by Kaplan–Meier analyses of DFS and OS, and then validated the results using the Molecular Taxonomy of Breast Cancer International Consortium (METABRIC) METABRIC cohort of 1904 breast cancer patients [30]. The values of the score for TCGA cohort had a unimodal Gaussian distribution and were significantly higher for tumor compared to adjacent normal tissue. In the US, approximately one-third of breast cancer patients have advanced disease at the time of diagnosis. We, therefore, chose to bin the patients into high- and low-score groups using the top one-third of the score as the cut-off. The hazard ratio (HR) for DFS with the 4-gene score was 1.74 (95% CI = 1.20–2.53; *p* < 0.001), which was higher than the HR with the individual genes (1.32–1.59; Figure 1A). Five-year DFS was 84.2% for the low-score group compared to 74.4% for the high-score group. As expected, the 4-gene score was also associated with overall survival (OS), with HR of 1.44 (95% CI = 1.22–1.69; *p* < 0.001; Figure 1A). Five-year OS rates were 86.5% and 72.0% for the low- and high-score groups. 

To validate the association of the 4-gene score with survival, we applied the scoring model to the METABRIC cohort. We analyzed disease-specific survival (DSS) instead of OS because OS data is not available for METABRIC. Patients with higher 4-gene scores had worse survival, mirroring the result for TCGA cohort (DFS: HR = 1.87, 95% CI = 1.35–2.58, *p* < 0.001; DSS: HR = 1.57, 95% CI = 1.32–1.85, *p* < 0.001; Figure 1A). Five-year DFS was 82.7% for the low-score group compared to 69.8% for the high-score group. Five-year DSS was 85.6% for the low-score group compared to 75.1% for the high-score group. Once again, the HR with the 4-gene score was higher than the HR with the individual genes (1.00–1.58; Figure 1A). Thus, the 4-gene score is an effective prognostic marker for breast cancer patients, with a value that is superior to the individual genes that constitute the score.

### 2.3. The 4-gene Score is An Independent Prognostic Factor for Breast Cancer Survival

We performed univariate and multivariate Cox regression analyses to evaluate if the prognostic value of the 4-gene score is independent of various other clinical and pathological factors. In univariate analysis of DFS in the TCGA cohort, cancer staging (III/IV vs. I/II), and the score (high vs. low) had significant HR (Table 1). For OS, age (≥50 vs. <50 years) and cancer subtype (triple-negative (TNBC) or HER2+ vs. hormonal receptor [HR]+/HER2−) were additional significant factors. In multivariate analyses using these significant factors, the score was found to be prognostic independently of other clinical factors for both DFS (HR = 1.91, 95% CI = 1.28–2.83; *p* = 0.001) and OS (HR = 2.18, 95% CI = 1.48–3.23; *p* < 0.001). The independent nature of the 4-gene score was also seen for the METABRIC cohort, for which subtype, stage, grade, and score were significant factors in univariate analyses of DFS and DSS (Table 1). In multivariate analyses with these three factors, HR for DSS was 1.34 (95% CI = 1.12–1.61; *p* = 0.002), and for DSS was 1.39 (95% CI = 1.15–1.67; *p* < 0.001).

### 2.4. The 4-Gene Score Correlates with Pathological Grade and Cancer Stage, but Also Prognosticates Survival within the Same Grade or Stage

We hypothesized that the 4-gene score is associated with aggressive clinical parameters. Accordingly, the score was significantly associated with Nottingham pathological grade in METABRIC cohort. Nottingham grades are unavailable for TCGA tumors. The 4-gene score was higher for advanced cancer staging in TCGA cohort and the trend was observed in METABRIC cohort (Figure 1B). When stratified, the score-based classifier was still a clinically and statistically significant prognostic model not only for the low-grade group (high vs. low score: DFS *p* = 0.005, DSS *p* = 0.015) but also the high-grade group (high vs. low score: DFS *p* = 0.008, DSS *p* = 0.004) (Figure 1C). In early-stage cancer (AJCC stage I and II), a high score was significantly associated with worse survival with OS in TCGA, and DFS and DSS in METABRIC cohort, although the score did not show a significant difference with DFS in TCGA. In advanced-stage cancer (stage III and IV), a high score was significantly associated with worse survival in both DFS and OS in TCGA, and in both DFS and DSS METABRIC cohorts. Thus, these results suggested that the 4-gene score may be able to add prognostic value particularly to Nottingham pathological grade and advanced cancer staging. 

### 2.5. The 4-Gene Score is Higher in TNBC Subtype, but Also Prognosticates Survival within Other Subtypes

Among the primary breast cancer patients of TCGA, the 4-gene score was higher in TNBC and HER2+ subtypes, which are known to be more clinically aggressive than HR+ and HER2− subtype (Figure 2). This result was validated with METABRIC cohort. Given that the score was higher in the aggressive subtype (TNBC), we asked if it detected patients with aggressive cancer within a subtype or if it merely detected aggressive subtypes. To this end, we performed survival analyses of TCGA cohort by cancer subtype and validated the results with METABRIC cohort. As shown in Figure 2, the survival analyses showed that the prognostic value of the 4-gene score was validated in the whole cohort and in the HR+/HER2− subtype alone. These observations suggest that high 4-gene scores did not reflect just the aggressive TNBC subtype but also the aggressiveness of tumors within the HR+/HER2− subtype.

### 2.6. Tumors with A High 4-Gene Score Have enriched Expression of Genes for Cancer Aggressiveness

To obtain biological insight into the nature of breast cancer that may explain the prognostic and predictive values of the 4-gene score, we conducted gene set enrichment analyses to compare tumors with high and low scores. Tumors with high scores had significant enrichment for 10 of the 50 mSigDb Hallmark gene sets (Figure 3A). The 4 sets related to cell cycle—E2F targets, G2M checkpoint, and MYC Targets v1 and v2—had enrichment in both TCGA and METABRIC cohorts. As increased cell proliferation is a marker of tumor aggressiveness, this suggests that tumors with high 4-gene scores are highly proliferative and more aggressive, which is consistent with the association between a high score and a higher Nottingham grade, advanced cancer stage, and worse survival (Figure 1 and Figure 2). Tumors with low scores had enrichment for only one gene set, early estrogen response, in both cohorts (Figure 3B). This is consistent with our finding that the 4-gene score is lower in HR+ cancer (Figure 1B). Given these results, we further hypothesized that the 4-gene score would associate with the classical cell proliferation marker Ki-67. Indeed, tumors with a high score had significantly higher gene expression for Ki-67 in both TCGA and METABRIC cohorts (Figure 3C). 

Recently there has been an accumulation of data that the tumor immune microenvironment plays a significant role in cancer biology and behavior [10,15,17,20]. Thus, it was of interest whether the 4-gene score or expression of any of the 4 genes was associated with regulatory T cell infiltration, a pro-cancerous immune cell. Neither the 4-gene score nor did the expression of any of the 4 genes correlate with FoxP3 gene expression, which is a surface marker of regulatory T cells (Appendix A).

### 2.7. Tumor 4-Gene Score is predictive of Distant Metastasis

Because our 4-gene score was developed based on the gene expression differences between the MDA-MB-231 and its lung-metastatic variant LM2-4, and because it is prognostic for survival in breast cancer in which distant metastasis is the major cause of death, we hypothesized that the 4-gene score is predictive of the development of distant metastasis. 

First, we examined the scores in a cohort of primary breast cancer tumors that eventually developed distant metastasis to bone alone, brain, or lung (GSE2603 [31]). We found that the 4-gene scores were significantly elevated in tumors that subsequently developed metastasis to the brain or lung, but not to bone alone, compared to the ones that did not develop any metastasis (Figure 4A). GSE2603 had a follow-up time of ≥3 years for patients with or without distant metastasis, which was deemed as the minimum time to confidently ascertain metastasis status. 

We also examined the 4-gene scores in primary and bone, brain, and lung metastasis tumors of the GSE110590 cohort [32]. Scores were significantly higher for brain and lung metastases compared to the primary tumor, whereas this difference was not seen for bone metastases. The findings depicted in Figure 4A indicate that not only primary breast cancers that spread to brain or lung but also the metastases have high 4-gene scores.

We used gene expression data of primary tumor and associated metastases of three previously published independent cohorts (GSE2603 [31], GSE2034 [33], and GSE12276 [34]) to analyze the association of the 4-gene score of the primary tumor with the time it took to develop metastasis at specific sites. We focused on bone, brain, and lung because they are the most common distant metastasis sites in breast cancer [35]. GSE2034 and GSE12276 cohorts had site-specific metastasis-free survival data for all three organs, whereas in GSE2603 cohort these data were available for only bone and lung (Figure 4B). Kaplan–Meier analyses of site-specific metastasis-free interval demonstrated that although the 4-gene score was not associated with bone metastasis in one of three cohorts, it was associated with brain metastasis in the other two (all logrank test *p* ≤ 0.033). For lung metastasis, the association was present in all three cohorts (all logrank test *p* ≤ 0.002; Figure 4B).

### 2.8. Effective Treatment Reduces Tumor 4-Gene Score

The association of a high 4-gene score with worse survival implicates that the score may reflect tumor aggressiveness. Thus, we hypothesized that the score would decrease with effective treatment that dampens the aggressiveness of tumors. In GSE28844 cohort, 33 patients were treated with a neoadjuvant chemotherapy (NAC) regimen consisting of anthracycline and taxane [36], and their clinical responses were measured with the Miller & Payne grading system as good (*n* = 8), mid (*n* = 14), and bad (*n* = 10). Indeed, the 4-gene score of tumors was significantly reduced after NAC for patients who had a good response (Figure 5A). Such a decrease was not seen for those who had an intermediate (mid) or poor (bad) response to treatment. A significant decrease in the 4-gene scores of tumors was also observed following successful hormonal therapy by anastrozole and fulvestrant in combination with gefitinib in the GSE33658 cohort (Figure 5A) [37].

### 2.9. Tumor 4-Fene Score is Predictive of Response to Neoadjuvant Chemotherapy

It is well known that highly proliferative, aggressive breast cancers are more likely to respond to NAC. We, therefore, speculated that a high 4-gene score prior to NAC will be associated with more cancer aggressiveness and thus a better response to NAC. To examine this, we analyzed each cancer subtype independently because it is well known that the therapeutic effect differs among subtypes [38,39,40]. In data from two independent studies (GSE23988 and GSE25066 [41], tumors with a high score prior to NAC were significantly associated with an increased likelihood of achieving pathological complete response (pCR) to treatment compared to patients with low score group especially in HR+/HER2- subtype (Figure 5B). These results suggest that the 4-gene score may have a predictive value for response to neoadjuvant treatment in HR+/ HER2 negative breast cancer. 

### 2.10. Tumors with A High 4-Gene score Have an Increased Expression of Immune Checkpoint Genes

Compared to tumors with low 4-gene scores, those with high scores had significantly increased expression of genes encoding immune checkpoint molecules such as *PD1*, (programmed death 1), and *PDL1* (programmed death ligand 1), *CTLA4* (cytotoxic T-lymphocyte-associated protein 4), *LAG3* (lymphocyte activation gene 3), *IDO1/2* (indoleamine dioxygenase 1/2) (Figure 5C). These results were also validated with the METABRIC cohort. A high 4-gene score was associated with increased expression of other immune checkpoint molecules as well; *HLA* (human leukocyte antigen), *TIGIT* (tyrosine-based inhibitory motif domain), *TIM-3* (T cell immunoglobulin domain and mucin domain 3), and *VISTA* (V-domain immunoglobulin-containing suppressor of T cell activation) (Appendix A). To this end, high 4-gene score tumors were associated with a higher level of immune checkpoint molecule expression, suggesting that such tumors will be more likely to respond to immune checkpoint inhibition. This result implicates that the 4-gene score may become a candidate for a patient selection biomarker for immune checkpoint inhibitor therapy.

## 3. Discussion 

The aim of our study was to develop a prognostic biomarker for occurrence of distant metastasis in breast cancer based on gene expression profiles of bulk tumors. To identify genes for such a biomarker, we compared the gene expression of poorly metastatic MDA-MB-231 human breast cancer cell line and its variant, LM2-4, which was selected in vivo after multiple rounds of implantation/resection for higher tendency for spontaneous metastatic spread [22,23]. 

The tumor expression of 4 genes that were upregulated in LM2-4 was identified to have a strong association with DFS in the TCGA cohort. We used these genes to develop a 4-gene score and found that the score was prognostic of survival for the TCGA patients, with higher scores associated with worse survival. These results were validated with another large cohort, METABRIC. Although we could not detect a clear difference in the performance in the TNBC and HER2+ breast cancers, the score was a prognostic biomarker for HR+/HER2- breast cancer patients. The result suggests that the score was not reflected by the aggressiveness determined by subtype alone. Furthermore, the prognostic performance of the 4-gene score was good for late-stage cancer. 

Interestingly, none of the genes that were included in the 4-gene score was previously considered to be a major player in breast cancer metastasis. PGF (placental growth factor) is a member of the pro-angiogenic vascular endothelial growth factor (VEGF) family, and its overexpression has been linked to pathological angiogenesis [42]. Thus, PGF was studied as a biomarker of cardiotoxicity by docetaxel and cyclophosphamide in breast cancer; however, its clinical trial has failed to demonstrate its utility [43]. Upregulation of PGF has also been found in human cervical squamous carcinoma, hemangioblastoma, melanoma, and meningioma, but not in breast cancer [44]. High PGF serum levels have been shown to be involved in the development of bevacizumab resistance in patients with metastatic colorectal cancer [45]. High expression of the PGF gene in gastric cancer was associated with an increased risk of lymph node metastasis and serosal invasion, and shorter survival [46]. PGF was reported to be associated with an angiogenesis pathway different from VEGF, and PGF blockade combined with the 5-FU + Folinic acid + Irinotecan (FOLFIRI) regimen improved the survival of patients with metastatic colorectal cancer who had resistance to chemotherapy [47]. PGF signaling was also reported to contribute to the anti-tumor effect of metformin in breast cancer [48]. PGF was also demonstrated to play a role in regulating the tumor immune microenvironment [49,50,51,52]. 

Downstream of tyrosine kinase (DOK) proteins are adaptor proteins that are widely implicated in the regulation of cell growth, differentiation, and death, as well as cell motility [53]. DOK4 is a positive regulator of the mitogen-activated protein kinase (MAPK) pathway [54], and it undergoes phosphorylation by the RET tyrosine kinase, whose over-expression is observed in estrogen receptor (ER)-positive breast cancer [55]. Studies have correlated RET gene expression with shorter metastasis-free survival and OS in breast cancer [56]. DOK4 was recently identified as a biomarker for poor prognostic outcomes in acute myeloid leukemia and involvement in nerve, breast, and other cancer cell lines [57]. DOK4 was also reported to be both up- and downregulated in non-small cell lung cancer with experiments pointing toward an epigenetic mechanism [58]. Indeed, DOK4 expression was increased by knockdown of miR-182 and miR-183 in Ductal Carcinoma in situ [59]. Finally, DOK4 was highly expressed in canine peripheral nerve sheath tumors that can be used to differentiate it from fibrosarcomas [60].

HCCS (holocytochrome c synthase) is an enzyme that is required for mitochondrial cytochrome c maturation and therefore respiration [61]. HCCS has been implicated in the molecular mechanisms that promote embryonic heart regeneration [62,63]. Inactivation of HCCS has been reported to result in respiratory chain deficiency and abnormal levels of cell proliferation, which lead to decreased pleural and peritoneal fold tissue and an increased risk of congenital diaphragmatic hernia [64,65,66]. HCCS mutation has been shown to be associated with the microphthalmia with linear skin defects or MIDAS (microphthalmia, dermal aplasia, and sclerocornea) syndrome, an X-linked disorder with male lethality [67,68,69]. HCCS is speculated to play a role in central nervous system development by modulating cell death and a link between mitochondrial dysfunction, intrinsic apoptosis, and developmental disorders [70,71]. To the best of our knowledge, there is no publication that reports an association of HCCS with cancer. 

SHCBP1 (SHC SH2-binding protein 1), a member of Src homolog and collagen homolog family, has been reported to be over-expressed in several malignancies including breast cancer. SHCBP1 regulates breast cancer cell proliferation through apoptosis and cell cycle regulation [72]. SHCBP1 expression was higher in breast cancer compared with adjusted tissue and it correlated with worse cancer stage and poor survival [72]. SHCBP1 was shown to promote gastric cancer [73]. SHCBP1 is upregulated in lung cancer cell lines and plays a role in apoptosis [74]. In lung cancer, SHCBP1 promotes cisplatin-induced apoptosis resistance, migration, and entry through activation of the Wnt pathway [75]. SHCBP1 levels in non-small cell lung cancer were inversely correlated with survival, demonstrating the potential of SHCBP1 as a prognostic biomarker in lung cancer [76]. Further studies using in vitro and in vivo systems are required to determine the functional roles of these genes for metastasis competency, and the value of the genes as therapeutic targets.

Gene expression measurements of primary breast cancer tumors have been investigated as predictive as well as prognostic biomarkers in hundreds of studies. At least 5 tumor gene expression-based biomarker assays are in clinical use. MammaPrint, the first such assay to be developed [77], relies on microarray-based measurement of expression of 70 genes. It is intended for prognosticating N0 stage I/II cancer with tumors of <5 cm. The Oncotype DX assay, which uses reverse transcription-PCR (RT-PCR) to quantify mRNAs of 21 genes in formalin-fixed tissue samples, was designed to predict the benefit of chemotherapy in N0 ER+ cancer treated with tamoxifen [78]. In the Prosigna test [79], 50 mRNAs are measured by RT-PCR for a score that can identify breast cancer subtype, and predict long-term distant recurrence in early stage, hormone receptor-positive cancer as well as pCR to chemotherapy. To our knowledge, neither these nor other previously published biomarkers contain any of the 4 genes that are used in our scoring method, and none is designed to be used to predict brain or lung metastasis.

Gene set enrichment analysis suggests that tumors with high 4-gene scores are more aggressive, as observed from their higher expression of genes involved in cells. On the other hand, tumors with low scores have a higher expression of genes associated with estrogen response. This is in agreement with the observation that HR+/HER2− tumors have low scores. The 4-gene score was prognostic for the development of distant metastasis. The scores of tumors that spread to the brain or lung were higher compared to those that did not metastasize or did so only to bone. Additionally, score values of brain or lung metastases were significantly higher compared to pre-metastatic primary tumors or bone metastases. The lack of predictive value of the score for bone metastasis may be because DFS was used for the selection of the genes to develop the 4-gene score, and metastases to bone are not as lethal as those to the brain or lung. We used three different cohorts and found that high score values of primary tumors were associated with shorter time-intervals for occurrence of metastasis, especially to the brain or lung.

The 4-gene scores of tumors were significantly reduced by chemotherapy or hormonal therapy when clinical response to treatment was good. Interestingly, the pCR rate for NAC among high-score tumors was significantly higher than low-score tumors. This observation was noted in two independent cohorts for the HR+/HER2− subtype. These results suggest that the 4-gene score may be useful for the patient selection for neoadjuvant treatment for HR+/HER2− breast cancer patients. The HR+/HER2− subtype accounts for approximately 70% of advanced breast cancer and is thus responsible for most of the deaths from the disease. It is believed that response to NAC, one of the main treatments for breast cancer, is relatively poor for HR+/HER2− breast cancer due to its low aggressiveness. While many efforts are ongoing to circumvent resistance mechanisms with a few strategies already incorporated into clinical practice, their effectiveness is still limited. Therefore, the ability to predict NAC responsiveness in HR+/HER2- breast cancer with a biomarker like our 4-gene score can be of significant utility.

Immune checkpoint inhibitors, such as those targeting the PD1/PDL1 axis, have shown modest activity as monotherapy in advanced breast cancer. Atezolizumab, an immune checkpoint inhibitor, combined with nab-paclitaxel was recently approved for breast cancer and now is in clinical use [80]. However, it remains a huge challenge to accurately identify patients who will respond to immunotherapy. It is therefore critical to find biomarkers for appropriate patient selection for immunotherapy. It has been reported that the expression levels of immune checkpoint genes are associated with prognosis of cancer patients [81]. In this study, we found that the 4-gene score is significantly associated with the expression level of many immune checkpoint genes in breast cancer tumors. These results suggest that the score, besides being a biomarker for response to chemotherapy or hormonal therapy, may also have a value in predicting response to immunotherapy in patients with breast cancer. 

Our study is not free from limitations. To assess the prognostic or predictive value of the 4-gene score, we used its within-cohort top one-third percentile value to stratify patients. Our study does not establish an absolute value for interpretation of the score. This is because we utilized data from a variety of studies for our analyses, and different gene expression and data processing methods were used in these studies. Further work is required to refine our scoring method (e.g., through the inclusion of housekeeping genes) so that the score can be interpreted by absolute value and used on a single-sample basis. Both hybridization microarray and RNA sequencing methods were utilized for gene expression profiling of tumors of the cohorts that we examined. How the gene expression measurement platform affects the 4-gene score value is unknown at this point and likely requires further investigation. Another limitation is that our 4-gene score was generated using only the upregulated genes in LM2-4 compared to parental MDA-MB-231 cells and disregarded the downregulated genes. Commonly, genes significantly downregulated are as informative as upregulated genes for building a score and omitting them may have missed novel biomarkers. Therefore, it is possible that a score that includes the downregulated genes may have better performance than the 4-gene score and development of such a score may become one of the future studies.

## 4. Material and Methods

### 4.1. Cell Culture

The MDA-MB-231 human breast cancer cell line (ExPASy Cellosaurus RRID: CVCL_0062) was acquired from ATCC, Manassas, VA. The LM2-4^LUC+^ variant of MDA-MB-231 [22] was kindly provided by Prof. Robert S. Kerbel of Sunnybrook Research Institute, University of Toronto, Canada. Both cell lines were cultured in RPMI-1640 medium (Thermo Fisher Scientific, Waltham, MA, USA) with 10% v/v fetal bovine serum (FBS; Peak Serum, Wellington, CO, USA) at 37 °C under a humidified atmosphere of 5% CO_2_. Identity of the cell lines was verified by simple tandem repeat profiling.

### 4.2. Gene Expression Profiling by RNA Sequencing

Each of the two cell lines was examined in triplicate. Both lines were cultured concurrently, and harvested at sub-confluence for RNA isolation on three different days. An affinity spin-column-based total RNA purification kit with a DNAse treatment step was used (Qiagen, Valencia, CA, USA). RNA preparations were quantified with TapeStation RNA ScreenTape (Agilent, Santa Clara, CA, USA). RNA integrity number values were above 8.0. Sequencing libraries were prepared from 1 µg RNA using reagents and protocols provided in TruSeq Stranded Total RNA Library Prep Gold kit (Illumina, San Diego, CA, USA). Ribosomal RNA depletion and 15 PCR cycles were employed during library preparation to increase the cDNA molarity. TapeStation D1000 ScreenTape (Agilent) and Library Quantification Kit (Kapa Biosystems, Wilmington, MA, USA) assays were used to confirm good quality of libraries. All six libraries were sequenced together on an Illumina HiSeq 2500 instrument using HiSeq Rapid Cluster Kit v2―Paired-End and Rapid SBS Kit v2 reagents to obtain paired reads of 100 bases. Casava software (version 1.8.2, Illumina Inc, San Diego, CA, USA) was used to demultiplex sequencing data. An average of 100 million sequence read-pairs were obtained for each sample. Raw read data were filtered to remove adapters and poor-quality segments, and then mapped in a splicing-aware manner to the hg38 human reference with TopHat2 [82]. TopHat2 is currently considered to be replaced by other software such as HISAT2 due to its inability to process modern RNA-sequence data; however, we believe TopHat2 still is suitable to process the current RNA-sequence data that were generated in 2017, when it was used in thousands of studies at that time. Gencode v25 gtf were used for correct read alignment across splice junctions [83]. Gene-level mapped read count values were obtained with HTSeq [84]. The read count data were normalized with DESeq2 [85] and log_2_-transformed for further analyses.

### 4.3. Gene Expression Analyses

For differential gene expression analyses of the cell lines, the DESeq2 Bioconductor package for R was used. Genes with expression changes of absolute log_2_ fold-change (FC) >1.2 and false discovery rate (FDR) <0.05 after adjustment for multi-testing by Benjamini–Hochberg method were considered as differentially expressed. For multiple testing adjustments, the Benjamini–Hochberg method was used to calculate the FDR, where the FDR < 0.05 was chosen as cut-off to identify the candidate genes. For gene set enrichment analysis, GSEA [6] software (http://software.broadinstitute.org/gsea/index.jsp) and mSigDb [86] Hallmark and C5 BP:GO biological process gene-set collections were used, and a nominal *p* value threshold of 0.05 and a FDR of 0.25, as recommended by the GSEA software, was used to deem significance.

### 4.4. Clinical and Gene Expression Data of Breast Cancer Patient Cohorts

Tumor gene expression and clinical data for 1094 patients of the TCGA breast cancer (BRCA) project [5] were obtained from cBioPortal [87] in August 2018. The RNA sequencing-based gene expression data in the portal had been processed and normalized with the RSEM method. Normalized microarray-based tumor gene expression data and clinical information for 1904 cases of the METABRIC cohort [30] were obtained from cBioPortal. In terms of survival data, disease-free survival (DFS) and overall survival (OS) were available in TCGA, and DFS and disease-specific survival (DSS) were available in METABRIC, but not OS. Clinicopathologic and normalized microarray-based gene expression data were also obtained for the studies of Wang et al. (GSE2034; *n* = 286) [33], Minn et al. (GSE2603; *n* = 82) [31], Siegel et al. (GSE110590; *n* = 16) [32], Bos et al. (GSE12276; *n* = 204) [34], Iwamoto et al. (GSE23988; *n* = 61) [88], Symmans et al. (GSE25066; *n* = 508) [41], Vera-Ramirez et al. (GSE28844; *n* = 33) [36], and Massarweh et al. (GSE33658; *n* = 11) [37], from the GEO repository. Due to the unavailability of tumor gene expression and relevant clinical data for a reasonable number of samples, only 4 cohorts (GSE2603, GSE110590, GSE2034, GSE12276) were used for the metastasis-related analyses. Similarly, only another 4 cohorts (GSE28844, GSE33658, GSE25066, GSE23988) were used for analyses of response to therapy. Extensive literature and gene expression repository searches were conducted to identify these cohorts. We included all cohorts with available data that we could find and did not exclude any cohort for the current study. As necessary, probe-level expression values were summarized using mean to obtain gene expression values. Log_2_-transformed data was used for further analyses.

### 4.5. Identification of genes highly expressed in LM2-4 cells and establishment of a 4-gene score

In order to determine which gene’s expression may be an independent prognostic marker for DFS, univariate Cox regression analysis was performed using the TCGA cohort. Among the 297 genes that were differentially upregulated in LM2-4 compared to parental MDA-MB-231 cells, *DOK4*, *HCCS*, *PGF*, and *SHCBP1* had HR ≥ 1.2 with *p* ≤ 0.02 in this analysis. Tumor expression of these four genes and DFS was then analyzed with a multivariate Cox regression model. The formula for calculating a 4-gene score using gene expression and Cox model regression coefficient values was thus established as:1.355 × (expression*^DOK4^*) + 1.641 × (expression*^HCCS^*) + 1.345 × (expression*^PGF^*) + 1.232 × (expression*^SHCBP1^*). Within any cohort, the high/low cutoff for the 4-gene score was defined as the top-third versus the bottom two-thirds throughout our work. Uniform cut-off values could not be used for all cohorts because of differences in platforms and data processing methods for generation of gene expression data.

### 4.6. Statistical Analyses 

All statistical analyses and data plotting were performed using R (http:///www.r-project.org/) or Microsoft Excel (version 16 for Windows, Redmond, WA, USA). Unless noted otherwise, Kaplan–Meier method with logrank test was used for survival analysis. A cut-off of 0.05 for *p* value was used to judge significance in statistical tests, which were always two-tailed and assumed equal group variances. 

## 5. Conclusions

In conclusion, by combining gene expression data of cancer cell lines with publicly available clinical and gene expression data of multiple published studies, we were able to establish a novel 4-gene score for breast cancer tumors. Multivariate and stratified analyses showed that the score is useful as a biomarker independent of clinical and pathological factors. The score may have utility for predicting survival, recurrence, and distant metastasis, as well as response to neoadjuvant therapy and possibly immunotherapy in breast cancer.

## Figures and Tables

**Figure 1 cancers-12-01148-f001:**
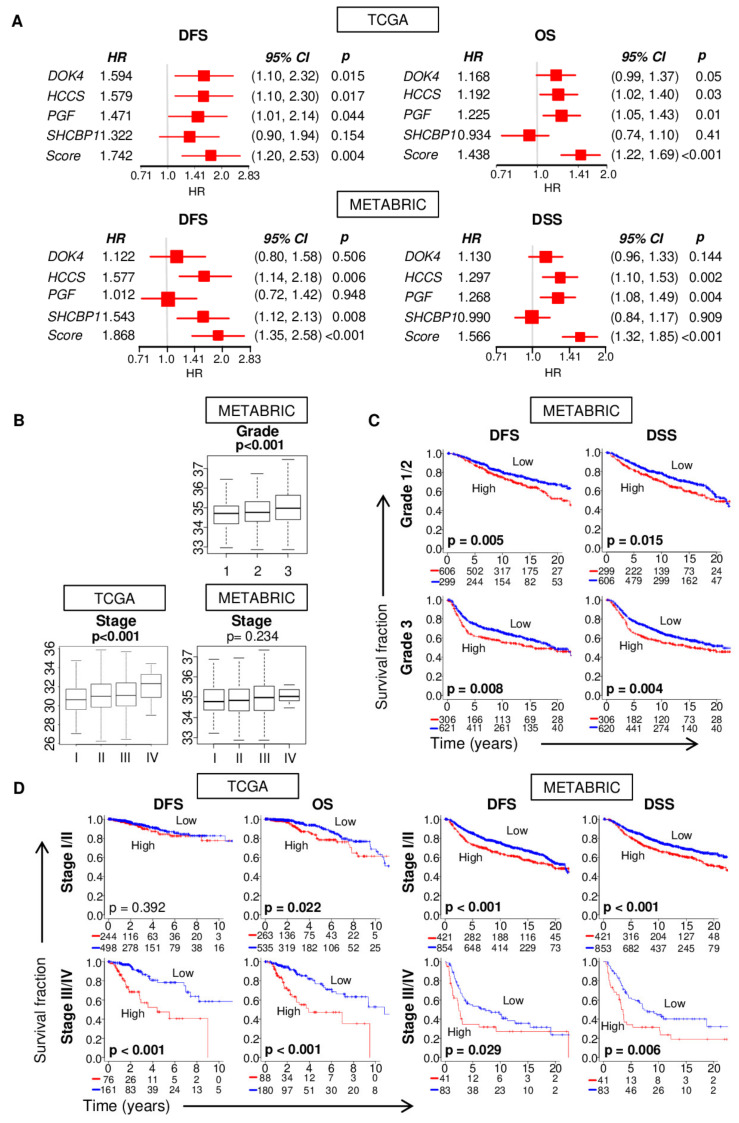
Patients with a high 4-gene score had poor survival and aggressive clinical parameters in two breast cancer cohorts. (**A**) Disease-free survival (*DFS*), and overall (*OS*) or disease-specific (*DSS*) survival in TCGA and METABRIC cohorts along with hazard ratios (*HR*) and their 95% confidence intervals and *p*-value are shown for the score and individually for each of its constituent genes. (**B**) Boxplots of 4-gene scores of tumors of different Nottingham pathological grades and American Joint Committee on Cancer (AJCC) stages. All plots are of Tukey type and boxes depict medians and inter-quartile ranges. *p* values were calculated with one-way ANOVA test. (**C**,**D**) Kaplan–Meier plots with logrank test *p* values are shown for association between the 4-gene score with DFS and OS or DSS for different grades and stages. Within-cohort one-third percentile value of the 4-gene score or expression of individual genes was used to classify patients into low and high groups.

**Figure 2 cancers-12-01148-f002:**
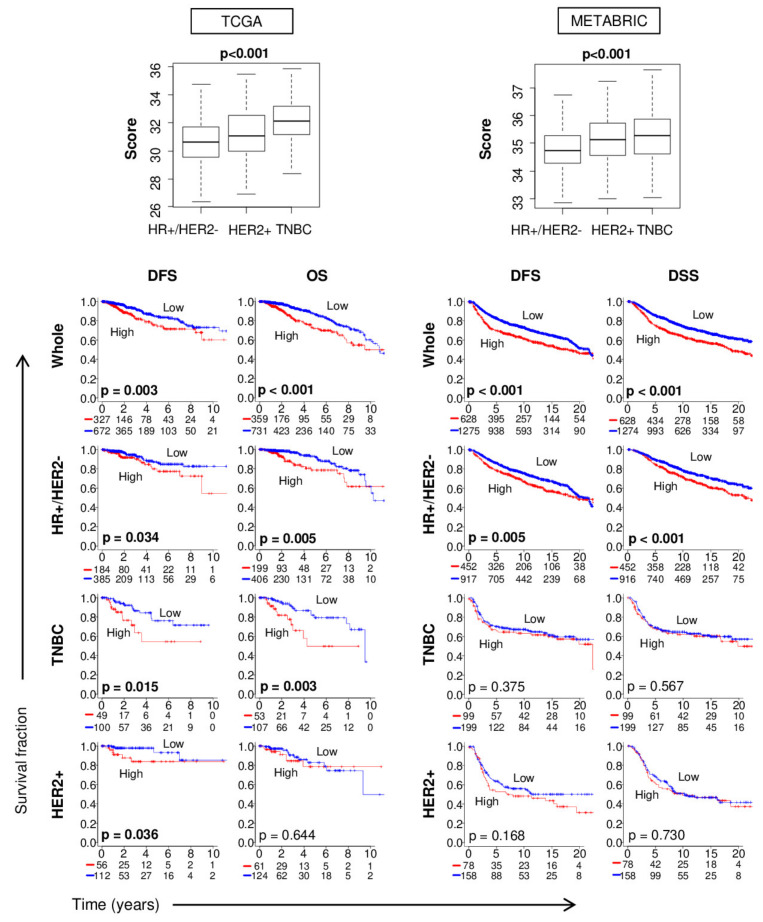
The 4-gene score is prognostic for disease-free survival (DFS) especially for hormonal receptor (HR)+/ human epidermal growth factor receptor 2 (HER2) cancer. At the top, boxplots show the association between the 4-gene score and cancer subtype. All boxplots are of Tukey type and boxes depict medians and inter-quartile ranges. *p* values were calculated with one-way ANOVA test. Below are Kaplan–Meier plots with logrank test *p* values for association between tumor 4-gene score and DFS and overall (OS) or disease-specific (DSS) survival for cancer of HR+/HER2−, HER2+, and triple-negative (TNBC) subtypes. Within-group one-third percentile value of the 4-gene score was used to divide patients into low and high classes.

**Figure 3 cancers-12-01148-f003:**
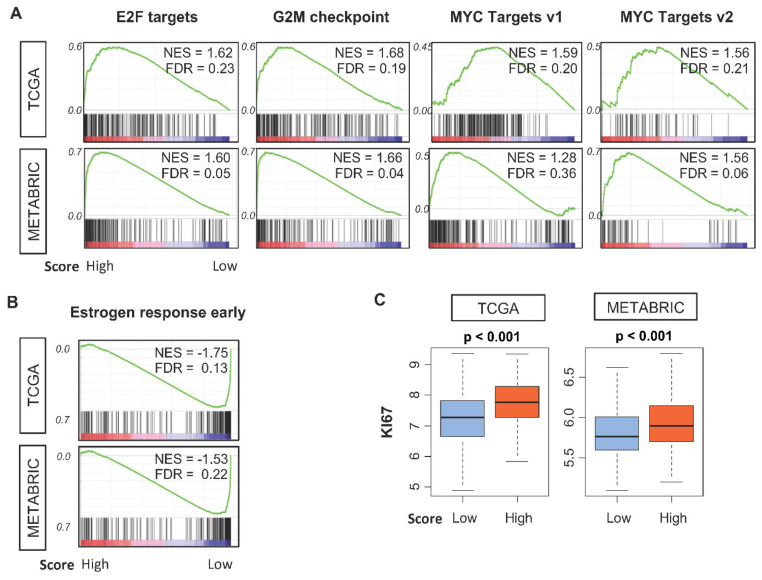
Gene expression enrichment in tumors of high or low 4-gene scores in TCGA (*n* = 1093) and METABRIC (*n* = 1903) cohorts. Enrichment plots along with normalized enrichment scores (*NES*) and false discovery rate (FDR) values are shown for the four and one gene sets whose expression was respectively enriched among high- (**A**) and low- (**B**) score tumors. **(C)** Tukey boxplots of tumor gene expression for KI67 gene (log_2_ transcripts per million). Boxes depict medians and inter-quartile ranges. *p* values were calculated with one-way ANOVA test. Within-group one-third percentile value of the 4-gene score was used to divide patients into low and high classes.

**Figure 4 cancers-12-01148-f004:**
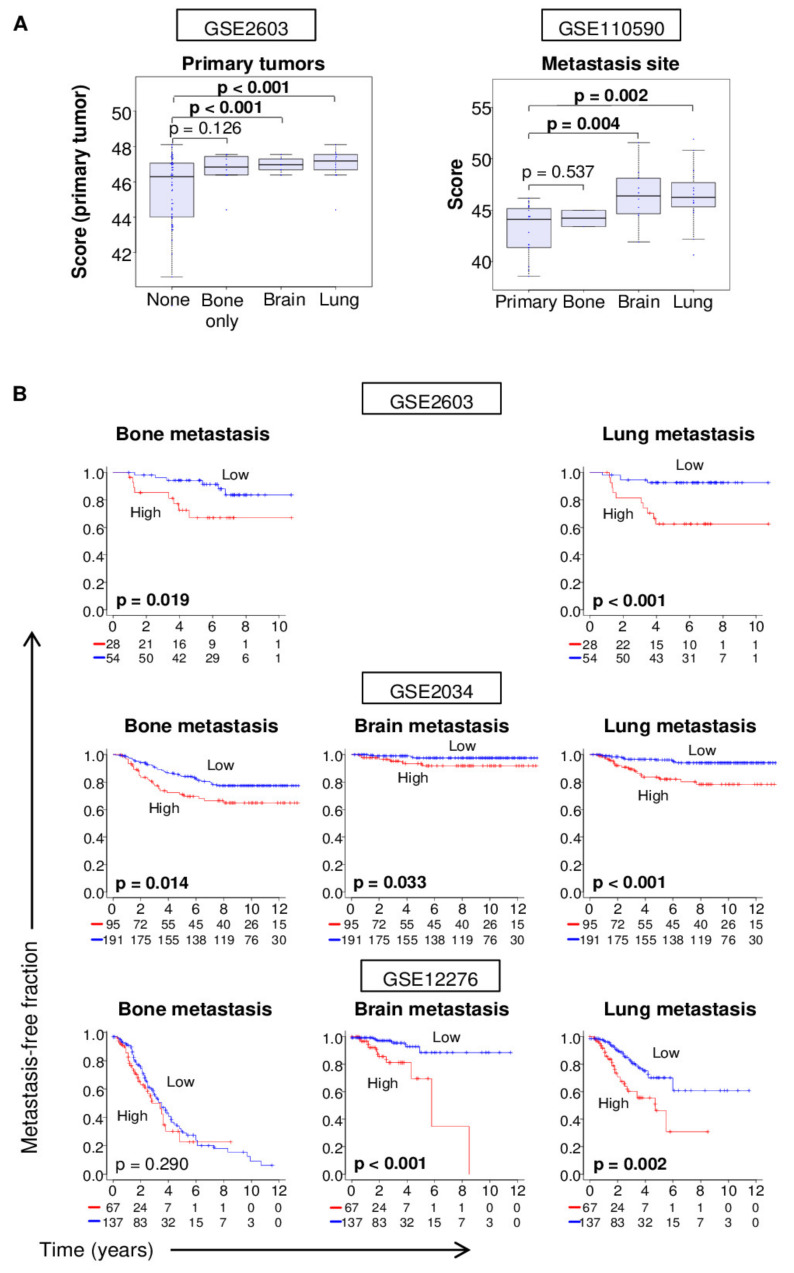
Association of the 4-gene score with metastasis in breast cancer. The examined cohorts are indicated. (**A**) Boxplots show 4-gene scores of primary tumors and metastases of patients who later developed metastasis to only bone, or to sites that included brain or lung, or did not have any metastasis. All boxplots are of Tukey type and the boxes depict medians and inter-quartile ranges. The depicted *p* values were calculated with standard t tests. (**B**) The Kaplan–Meier survival plots depict metastasis-free survival for metastasis to bone, brain, or lung based on the pre-metastasis primary tumor’s 4-gene score for three cohorts. For each plot, within-cohort one-third percentile value of the score was used to classify tumors into low and high groups.

**Figure 5 cancers-12-01148-f005:**
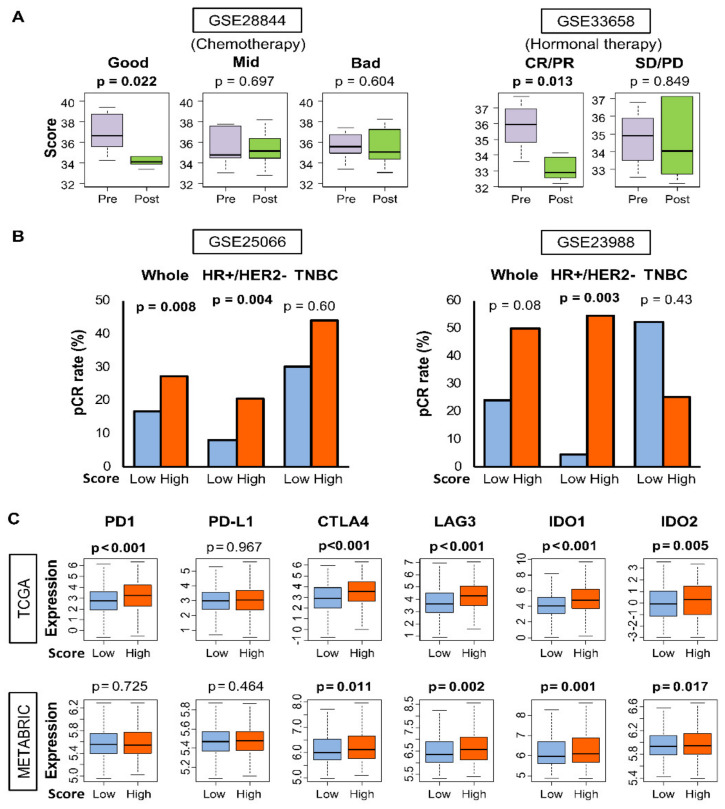
Relationships between response to therapy and the gene score. The examined cohorts are indicated. (**A**) Boxplots show the scores of pre- and post-neoadjuvant chemotherapy tumors of GSE28844 cohorts, and of pre- and post-therapy tumors of GSE33658 cohort in which neoadjuvant hormonal therapy was used. For GSE28844, response categories are as per Miller–Payne criteria. RECIST criteria are used for GSE33658 (*PD*, partial disease; *SD*, stable disease; *PR*, partial response; *CR*, complete response). Standard t tests were used to determine *p* values. All boxplots are of Tukey type and the boxes depict medians and inter-quartile ranges. (**B**) Barplots show the pathologic complete response (*pCR*) rates in high- and low-score groups of GSE25066 (*n* = 508) and GSE23988 (*n* = 61). Fisher’s exact test was used to determine *p* values. The one-third percentile value of the 4-gene score was used to classify tumors into low and high groups. (**C**) Comparisons of gene expression (log_2_ transcripts per million) of immune response genes in tumors of high or low 4-gene scores in both TCGA and METABRIC cohorts. Boxplots of expression of immune response genes in tumors with high and low 4-gene scores. The one-third percentile value of the 4-gene score was used to classify tumors into low and high groups. All boxplots are of Tukey type and the boxes depict medians and inter-quartile ranges. The depicted *p* values were calculated with standard t tests. *CTLA4*, cytotoxic T-lymphocyte-associated protein 4; *IDO1/2*, indoleamine dioxygenase 1/2; *LAG3*, lymphocyte activation gene 3; *PD-1*, programmed death-1; *PD-L1*, (programmed death ligand 1).

**Table 1 cancers-12-01148-t001:** Survival analyses of 4-gene score and other factors in TCGA and METABRIC cohorts.

Factors	Univariate		Multivariate	
	HR (95% CI)	*p*	HR (95% CI)	*p*
TCGA (DFS)				
Age (≥ 50 vs. < 50 y)	0.91 (0.67–1.33)	0.609		
Subtype (HR+/HER2− vs. other)	0.67 (0.44–1.01)	0.054		
AJCC stage (III/IV vs. I/II)	2.86 (1.93–4.21)	<0.001	2.89 (1.96–4.27)	<0.001
Race (white vs. black)	0.66 (0.42–1.02)	0.059		
Histology (ILC vs. IDC)	1.15 (0.71–1.84)	0.577		
Score (high vs. low)	1.74 (1.20–2.53)	0.004	1.91 (1.28–2.83)	0.001
TCGA (OS)				
Age (≥ 50 vs. < 50 y)	1.53 (1.06–2.22)	0.024	1.81 (1.16–2.83)	0.009
Subtype (HR+/HER2− vs. Other)	0.611 (0.42–0.89)	0.010	0.68 (0.46–1.00)	0.052
AJCC stage (III/IV vs. I/II)	2.64 (1.89–3.69)	<0.001	3.18 (2.18–4.65)	<0.001
Race (white vs. black)	0.82 (0.55–1.23)	0.347		
Histology (ILC vs. IDC)	0.90 (0.58–1.40)	0.637		
Score (High vs. Low)	1.87 (1.35–2.58)	<0.001	2.18 (1.48–3.23)	<0.001
METABRIC (DFS)				
Age (≥ 50 vs. < 50 y)	1.12 (0.94–1.33)	0.205		
Subtype (HR+/HER2- vs. Other)	0.64 (0.55–0.75)	<0.001	0.75 (0.62–0.91)	0.004
AJCC stage (III/IV vs. I/II)	3.03 (2.37–3.88)	<0.001	2.84 (2.21–3.64)	<0.001
Cellularity (high vs. other)	1.02 (0.87–1.18)	0.840		
Histology (ILC vs. IDC)	0.97 (0.73–1.30)	0.862		
Grade (3 vs. 1/2)	1.52 (1.30–1.77)	<0.001	1.55 (1.04–2.32)	0.031
Score (High vs. Low)	1.44 (1.23–1.68)	<0.001	1.34 (1.12–1.61)	0.002
METABRIC (DSS)				
Age (≥ 50 vs. < 50 y)	1.17 (0.98–1.39)	0.070		
Subtype (HR+/HER2− vs. Other)	0.67 (0.57–0.78)	<0.001	0.78 (0.65–0.95)	0.014
AJCC stage (III/IV vs. I/II)	2.94 (2.30–3.76)	<0.001	2.77 (2.15–3.55)	<0.001
Cellularity (high vs. other)	1.01 (0.87–1.18)	0.868		
Histology (ILC vs. IDC)	0.98 (0.74–1.31)	0.900		
Grade (3 vs.1/2)	1.49 (1.27–1.74)	<0.001	1.57 (1.05–2.34)	0.027
Score (High vs. Low)	1.46 (1.25–1.71)	<0.001	1.39 (1.15–1.67)	<0.001

AJCC, American Joint Committee on Cancer; CI, confidence interval; DSS, disease-specific survival; DFS, disease-free survival; HR, hazard rate; HR+/HER2-, hormonal receptor positive/human epidermal growth factor receptor 2 negative; ILC, infiltrating lobular carcinoma; IDC, infiltrating ductal carcinoma; OS, overall survival; TNBC, triple negative breast cancer.

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
