# Peer review of "A Novel 4-gene Score to Predict Survival, Distant Metastasis and Response to Neoadjuvant Therapy in Breast Cancer"

_cancers, 2020, doi:10.3390/cancers12051148_

Round 1

Reviewer 1 Report

  1. The authors compared gene expression profiles of MDA-MB-231 (human breast cancer cell line) and LM2-4 (lung met derivative) and identified 250 genes with reduced expression and 297 with increased expression. Of these 297 genes, four (up-regulated genes: DOK4, HCCS, PGF, and SHCBP1) were statistically significantly associated (individually and using a score cutoff) with DFS in the TCGA population. This was true in a multivariable setting as well as in the overall survival setting using TCGA data. METABRIC data were used to validate these results using DFS and DSS. The authors note on page 7 that "These observations suggest that high 4-gene scores did not reflect just the aggressive TNBC subtype but rather the aggressiveness of tumors within each subtype." This is not supported by the results, particularly for TNBC and HER2+, neither of which were validated in METABRIC. These results should be discussed in more detail.
  2. On page 3, line 103 (and for the majority of the manuscript), the authors note that the definition of high/low cutoff for the 4-gene score was defined as the top third versus the bottom two thirds. However, in the methodology section (p 16, line 441), this cutoff is defined as the top 25% versus the bottom 75%. This needs clarification, obviously, but also were multiple cutoffs considered and if so, were the results consistent?
  3. It is noted on page 3, line 111 that "OS data was not available" for METABRIC. This is not explicit throughout the manuscript and sometimes directly contradicted. For example, in the abstract it is noted that there is an association between the 4-gene score and survival in METABRIC. Also, on line 100, "with survival data availability".
  4. Survival figure numbers at risk are difficult to read and it is unclear which line is attributed to which group.
  5. The abstract should be more measured in its message. There are some exciting results that are to be conveyed, however, there are particular subgroups to which this does not apply. Furthermore, it is unfair to claim "In examination of 4,199 patients of 10 other cohorts..." for many reasons. First, these additional analyses were in fact in only a minority of 4,199 patients. For example, the mets analyses were in 14% of the possible 4,199. This should be explicit in the abstract as well as the figures themselves, and not just in the figure descriptions. Similar comments apply to the response to therapy results which were investigated in even fewer patients.
  6. Why weren't all studies included in the mets analyses and the response to therapy analyses? Were data unavailable?
  7. Was a competing risks time to mets model considered to investigate the association between the 4-gene score with mets?

Reviewer 2 Report

Concerns to Address to Manuscript:

Methods & Experimental Design

Line 403: Why PCR amplification for library prep?

Line 410: Did the authors use the gene model for correct read alignment across splice junctions? If yes, indicate which annotation/gene model was used. If no, please explain the rationale for not correcting for splice junctions and how the confounding pseudogene count bias for actual gene count was mitigated.

Also, the authors used a first generation aligner, TopHat2, which is considered obsolete due its inability to process modern RNA-seq data. What is the rationale for using TopHat2?

Line 416: What is the q-value cutoff for BH correction?

Results, Discussion, & Conclusions Concerns

Line 436: Why only upregulated genes were used in Cox regression analysis? Genes significantly downregulated in LM2-4 are as informative for building this score. Omitting the downregulated genes bias the identification process for novel biomarkers for poor prognosis, leading to false hits. Address this shortcoming in the methods, its impact upon the data, and its interpretation and application to patients by clinicians. Can expression of these 4-genes be used by physicians to predict patients with poor prognosis?

Importantly in the discussion there is a lack of gene or biochemical description, research reports, or literature on the 4 identified genes, DOK4, HCCS, PGF and SHCBP1, in the context of cancer drivers, passengers, phenotypes or outcomes.  Incorporate an overview in the Discussion on each of the 4 genes, focusing on cancer, cell biology, or developmental biology, if known. For example, DOK4 was identified a year ago as a biomarker for poor prognostic outcomes in leukemia and involvement in nerve, breast, and other cancer cell lines (Zhang e tal 2019 ). Conduct an extensive literature search on PubMed on the other 3 genes.

1: Zhang L, Li R, Hu K, Dai Y, Pang Y, Jiao Y, Liu Y, Cui L, Shi J, Cheng Z, Fu L. Prognostic role of DOK family adapters in acute myeloid leukemia. Cancer Gene  Ther. 2019 Sep;26(9-10):305-312. doi:  0.1038/s41417-018-0052-z.

2: 4: Hannafon BN, Sebastiani P, de las Morenas A, Lu J, Rosenberg CL. Expression of microRNA and their gene targets are dysregulated in preinvasive breast cancer. Breast Cancer Res. 2011 Mar 4;13(2):R24. doi: 10.1186/bcr2839.

3: Klopfleisch R, Meyer A, Lenze D, Hummel M, Gruber AD. Canine cutaneous peripheral nerve sheath tumours versus fibrosarcomas can be differentiated by neuroectodermal marker genes in their transcriptome. J Comp Pathol. 2013 Feb;148(2-3):197-205. doi: 10.1016/j.jcpa.2012.06.004.

4: Gray SG, Al-Sarraf N, Baird AM, Gately K, McGovern E, O'Byrne KJ. Transcriptional regulation of IRS5/DOK4 expression in non-small-cell lung cancer  cells. Clin Lung Cancer. 2008 Nov;9(6):367-74. doi: 10.3816/CLC.2008.n.053.

Multiple minor grammar mistakes to correct throughout the manuscript:

For instance, the word “data” is a plural noun and proper verb conjugation requires revision throughout the entire text. Revise text to “data were analyzed” from incorrect “data was analyzed”.

Reviewer 3 Report

The manuscript titled “A novel 4-gene score to predict survival, distant metastasis, and response to neoadjuvant therapy in breast cancer” investigated on possible prognostic biomarkers for breast cancer metastasis and progression. Comparing gene expression of two different breast cancer cell lines (metastatic versus non-metastatic), the authors identified four genes up-regulated in non-metastatic cell line. This gene signature associated with disease-free survival, overall survival and predict metastasis in brain and lung. Finally, tumors with high score had increased expression of T cell exhaustion marker genes.

The manuscript is well written and the data supported the conclusion. The authors based their study on the gene expression metastatic and non-metastatic breast cancer cell lines in order to find a gene signature able to trace cancer progression. The idea is quite convincing and the study has potential applicative results. However, some concerns limited the strength of the message.

Specifically, the authors analyzed MDA-MB-231 and its metastatic variant LM2-4 breast cancer cell-line. A detailed characterization (i.e in terms of age, number of passages in vitro and SC-like properties) of these specific “laboratory cells” is necessary to better understand their phenotype. Moreover, could be important to explore whether the observed gene signature is confirmed on primary tumour cells obtained from breast cancer tissue.

The authors found that tumors with a high 4-gene score have an increased expression of immune checkpoint genes. The authors showed that cancers with high scores had significantly increased expression of several genes encoding immune checkpoint molecules, such as CTLA4 IDO, LAG3, PD1, PDL1 TIGIT and VISTA. However, in the Fig. 5 are reported only some inhibitor checkpoint molecules but in the legend (and in the main text) were mentioned most of them. Please, clarify and explain better these findings.

It could be interesting analyze in the different cohorts whether identified genes also associated with molecules associate with regulatory T cell, for example FoxP3 gene.

Round 2

Reviewer 2 Report

    I appreciate the time and effort the team has taken to address all the concerns by all of the reviewers, with new analysis, more detailed methods, and substantially more citations that support the results and conclusions.

    For the future, I encourage the bioinformatics team members to mine and model the downregulated genes as relevant for clinical outcomes because as a loss, or lack, of expression of tumor suppressors can reveal the molecular drivers in cancer, such as in Li-Fraumeni syndrome and the loss of TP53 for example. I look forward to a second manuscript on those data.
